

# Validating the water vapour content from a reanalysis product and a regional climate model over Europe based on GNSS observations

Julie Berckmans[1,2], Roeland Van Malderen[1], Eric Pottiaux[3], Rosa Pacione[4], and Rafiq Hamdi[1]

[1]Royal Meteorological Institute, Ringlaan 3, 1180 Brussels, Belgium
[2]Centre of Excellence Plants and Ecosystems (PLECO), University of Antwerp, Belgium
[3]Royal Observatory of Belgium, Ringlaan 3, 1180 Brussels, Belgium
[4]e-GEOS S.p.A. ASI/CGS Matera, Italy

*Correspondence to:* Julie Berckmans (julie.berckmans@vito.be)

**Abstract.** The use of ground-based observations is suitable for the assessment of atmospheric water vapour in climate models. Global Navigation Satellite Systems (GNSS) provide information on the Integrated Water Vapour (IWV), at a high temporal and spatial resolution. We used IWV observations at 100 European GNSS sites to evaluate the regional climate model ALARO running at 20 km horizontal resolution and coupled to the land surface model SURFEX, driven by the European Centre for Medium-Range Weather Forecasts (ECMWF) Interim Re-Analysis (ERA-Interim) data. The observations recorded in the selected stations span from 1996 to 2014 (with minimum 10 years) and were homogeneously reprocessed during the second reprocessing campaign of the EUREF Permanent Network (EPN Repro2). The outcome of the reprocessing was then used to compute IWV time series at these stations. The yearly cycle of the IWV for the 19-year period from 1996 to 2014 reveals that the model simulates well the seasonal variation. Although the model overestimates IWV during winter and spring, it is consistent with the driving field of ERA-Interim. However, the agreement with ERA-Interim is less in summer, when the model demonstrates an underestimation of the IWV. The model presents a cold and dry bias in summer that feedbacks to a lower evapotranspiration and results in too few water vapour. The spatial variability among the sites is high and shows a dependence on the altitude of the stations which is strongest in summer and by ALARO-SURFEX. The IWV diurnal cycle presents best results with ERA-Interim in the morning, whereas ALARO-SURFEX presents best results at midnight.

## 1 Introduction

The Earth's climate is controlled by greenhouse gases. Atmospheric water vapour is the most prominent greenhouse gas and plays a key role in the feedback process of a changing climate (Cess et al., 1990). An increase in temperature enhances the water-holding capacity of the atmosphere, thus amplifying the greenhouse effect (Trenberth, 1998) Therefore, a good simulation of the water vapour by climate models is crucial. However, the common measuring systems for water vapour observations, such as radiosondes and satellite's instruments, are not adequate for the validation of climate models (Vey et al., 2010).

The use of ground-based observations has increasingly attracted the attention for measuring water vapour, because of its high temporal and spatial variability. The ground-based measurements of Global Navigation Satellite System (GNSS) sig-



nals provide information on the water vapour integrated over the total column of the atmosphere (Wang et al., 2005). They have the additional advantage over traditional systems, such as radiosondes, of measuring during all weather conditions and at high spatial and temporal resolutions. The most common GNSS is the American Global Positioning System (GPS), but additional systems exist, such as the Russian Global Navigation Satellite System (GLONASS), the Chinese Beidou system, or

the European Galileo.

The signals received from GNSS are delayed by the neutral atmosphere, consisting of the troposphere, stratosphere and mesosphere. The delay induced by the lower neutral atmosphere on the GNSS signal is a valuable source of information for the estimation of the amount of atmospheric water vapour. This delay is referred to as the tropospheric delay, and can be converted into IWV by using the surface pressure and a water vapour weighted mean temperature (Tralli and Lichten, 1990).

GNSS receivers demonstrate a high stability, as they monitor the water vapour under almost all weather conditions, with a temporal resolution of one hour over a long period. Therefore, these long-term measurements are highly suitable for climate model validation (Wang and Zhang, 2009).

Although considerable research has been devoted to climate monitoring by considering trends in IWV retrieved by GNSS (Gradinarsky et al., 2002; Nilsson and Elgered, 2008; Ning and Elgered, 2012), less attention has been paid to climate model

validation. Ning et al. (2013) assessed a Regional Climate Model (RCM) for IWV and compared it to GPS measurements. They revealed an underestimation of the IWV, due to a cold and dry bias represented by the model. Sites close to the sea resulted in larger mean IWV differences. Therefore, some authors suggested to be careful when validating model grid boxes with point observations (Ning et al., 2013).

Challenges arise when attempting to explain the temporal and spatial variability of the IWV differences between a RCM

and GNSS observations (Guerova et al., 2016). The GNSS observations that serve as the basis for this study arrive from a network within the International Association of Geodesy Regional Reference Frame sub-commission for Europe (EUREF, http://epncb.oma.be). The efforts made during the COST Action ES1206 (GNSS4SWEC) have stimulated and supported the production of homogeneously reprocessed long-term datasets of tropospheric delays, such as the second tropospheric repro-cessing campaign carried out within EUREF (EPN Repro2, Pacione et al., 2017). This dataset (converted into IWV) allows

us to validate the regional distribution of water vapour from climate models. We evaluate the ability to simulate the IWV by the RCM ALARO-0 (Gerard et al., 2009) coupled to SURFEXv5 (Masson et al., 2013), driven by the European Centre for Medium-Range Weather Forecasts (ECMWF) Interim Re-Analysis (ERA-Interim, Dee et al., 2011), at 100 European GNSS sites for a 19-year period from 1996 to 2014. Both ERA-Interim and ALARO-SURFEX have not assimilated GNSS informa-tion, so the comparisons among both with the GNSS observations are fair. Specifically, the inter-annual, seasonal and diurnal

variations are investigated, as well as the spatial variation.

This manuscript is organised as follows: in Sect. 2 we describe the procedure of the IWV datasets and the methodology. Sect. 3 covers the results on the different scales of variability. The results are concluded in Sect. 4.



## 2   Datasets

### 2.1   GNSS-derived IWV dataset

When travelling through the neutral atmosphere (hereafter referred to as troposphere), the signals emitted by GNSS satellites and received by ground-based receivers undergo a propagation delay called the tropospheric delay. This is the difference be-

tween the actual path that the signal travels through the troposphere and the path that the signal would have followed in vacuum conditions. During the GNSS data processing, this path delay is estimated in the zenith direction and is referred to as Zenith Total Delay (ZTD). A large infrastructure of 323 continuously operating GNSS stations is maintained by EUREF to form the EUREF Permanent Network (Bruyninx et al., 2012; Ihde et al., 2013). Within EUREF, the GNSS observations from the EPN are analysed operationally by several analysis centres to determine precisely the coordinates of the stations. In the process

they also estimate the ZTD at these stations. Over time, the computation carried out by the different EUREF analysis centres experiences sometimes inconsistencies due to e.g. the updates of the reference frame and applied models, the use of difference elevation cut-off angles, different mapping functions (Ning et al., 2016a). A homogeneous reprocessing of the whole dataset is thus mandatory to overcome such problems prior using this dataset for any climate application. Within EUREF, the GNSS observations from the entire EPN and for the 19-year period (1996-2014) have been carefully and homogeneously reprocessed

during the so-called second EPN reprocessing campaign (EPN-Repro2, Pacione et al., 2017) in order to produce a climate-quality tropospheric dataset over Europe.

The ZTD consists of a Zenith Hydrostatic Delay (ZHD) component that reflects the whole density of the neutral atmosphere and a Zenith Wet Delay (ZWD) component that reflects the wet state of the atmosphere caused by water vapour. For climate

application, ZWD is usually converted to IWV, and expressed as kg m$^{-2}$ as it is the mass of water vapour in a column of the atmosphere. The conversion is done by subtracting the hydrostatic delay from the estimated ZTD:

$$ZWD = ZTD - ZHD \tag{1}$$

$$IWV = \kappa.ZWD \tag{2}$$

According to Askne and Nordius (1987) and Hogg et al. (1981) the coefficient $\kappa$ is:

$$\kappa = \frac{10^6}{\rho_{\mathrm{w}} R_{\mathrm{v}} (\frac{k_3}{T_{\mathrm{m}}} + k'_2)} \tag{3}$$

where $\rho_{\mathrm{w}}$ is the density of liquid water (999.97 kg m$^{-3}$), $R_v$ is the specific gas constant of water vapour (461.51 J$^{-1}$ kg$^{-1}$), $T_m$ is the vertically integrated mean temperature within an atmospheric water vapour column represented by N levels, where the temperature at each level is weighted with the specific humidity. Besides, $k_3$ and $k'_2$ are atmospheric refractivity constants





(3739 $\pm$ 12 K$^2$ h Pa$^{-1}$ and 0.22 $\pm$ 0.02 K$^2$ h Pa$^{-1}$). In this work, $T_m$ was derived from the ERA-Interim pressure level data. This is an appropriate strategy as discussed by Hagemann and Bengtsson (2003) and is suitable for this study.

Following (Bevis et al., 1992) the ZHD can be written as a function of the pressure measured at the GNSS station:

$$5 \quad ZHD = 0.0022768 \frac{P_{\text{GNSS}}}{f(\lambda, H_{\text{GNSS}})} \qquad (4)$$

where $P_{GNSS}$ is the pressure at the GNSS antenna height (in hPa) and was derived from the ERA-Interim surface pressure, but converted to the antenna height by making use of the barometric formula. The $f(\lambda, H_{GNSS})$ is a function close to unity that accounts for the variation in gravitational acceleration, with latitude $\lambda$ and height $H$ of the surface above the ellipsoid (in m) (Saastamoinen, 1972):

$$10 \quad f(\lambda, H_{\text{GPS}}) = 1 - 0.00266 cos(2\lambda) - 0.00000028 H_{\text{GNSS}} \qquad (5)$$

The conversion of ZTD to IWV introduces errors that affect the uncertainty of the IWV estimation. The pressure and mean temperature derived from ERA-Interim influence the accuracy of the ZHD and $\kappa$ coefficient (Ning et al., 2013, 2016b). The uncertainties in ZTD determine 75 % of the total IWV uncertainty for three sites in Germany, New-Zealand and Norway from the Global Climate Observing System Reference Upper-Air Network (Ning et al., 2016a; Bodeker et al., 2016). For these sites, the total IWV uncertainty was 0.68, 0.65 and 0.53 kg m$^{-2}$ for the site in Germany, New-Zealand and Norway respectively. The corresponding standard deviations that represent the variations in IWV uncertainty were 0.16, 0.13 and 0.06 kg m$^{-2}$ respectively (Ning et al., 2016a).

## 2.2 ERA-Interim IWV dataset

ERA-Interim is a global atmospheric reanalysis from 1979, continuously updated in real time, but with a 2006 release of the data assimilation system. ERA-Interim does not incorporate the assimilation of GNSS data. The system includes a 4-dimensional variational analysis (4D-Var) with a 12-hour analysis window, giving a temporal resolution of 6h. The spatial resolution of the dataset is approximately 80 km (0.75° × 0.75°) on 60 vertical levels from the surface up to 0.1 hPa. Today, this is the most state-of-the-art reanalysis product of the ECMWF, but an improved reanalysis product called ERA5 is currently under development. Unfortunately, this product was not available during this study so we were appointed to make use of ERA-Interim.

The ERA-Interim IWV values have been extracted from the surface fields at the GNSS station location. The IWV values from the four grid boxes surrounding the GNSS station were horizontally interpolated, and weighted with the inverse distance to the GNSS station. Furthermore, the ERA-Interim IWV was corrected using a vertical interpolation to the GNSS station height, as the model topography using in ERA-Interim usually does not agree with the height of the GNSS station height. We



applied the approach proposed by Hagemann and Bengtsson (2003), in which the adjustment is obtained by the (numerical) integration of the specific humidity q over the height difference between the GNSS station and the model surface in 30-m steps, which generally corresponds to a pressure difference smaller than 4 hPa. A constant dewpoint depression is assumed to height-adjust the specific humidity.

## 2.3 ALARO-SURFEX IWV dataset

The regional climate model used in this study is the ALARO model version 0, a configuration of the Aire Limitée Adaptation Dynamique Développement International (ALADIN) model with improved physical parameterisations (Gerard et al., 2009), combined with the Application de la Recherche à l'Opérationnel à Meso-Echelle (AROME), first baseline version released in 1998. The ALADIN model is the limited area model version of the global scale Action de Recherche Petite Echelle Grande Echelle Integrated Forecast system (ARPEGE-IFS) (Bubnovà et al., 1995; ALADIN International Team, 1997). ARPEGE is a global spectral model, with a Gaussian grid for the grid-point calculation. The vertical discretisation uses hybrid terrain-following pressure coordinates. The ALARO-0 model has been developed with the ARPEGE Calcul Radiatif Avec Nebulosité (ACRANEB) scheme for radiation based on Ritter and Geleyn (1992). This ALARO-0 model configuration has been operating at the Royal Meteorological Institute of Belgium (RMI) for its operational numerical weather forecasts since 2010. The new physical parameterisation within the ALARO-0 model was specifically designed to run at convection-permitting scales, with a particular focus on an improved convection and cloud scheme (Gerard and Geleyn, 2005; Gerard, 2007; Gerard et al., 2009). The ALARO-0 model domain is centred over Western Europe at 46.47°N and 2.58°E with a dimension of 149 x 149 horizontal grid points and spacing of 20 km in both horizontal axes, with a Lambert conformal projection (Fig. 1). The model consists of 46 vertical layers with the lowest model level at 17 m and the model top extending up to 72 km. The regional climate model is coupled to the land surface model SURFace EXternalisée (SURFEX, Masson et al., 2013). SURFEX is based on a tiling approach with each tile providing information on the surface fluxes according to the type of surface: nature, town, inland water and sea.

The regional climate model was driven by initial and lateral boundary conditions provided by the ERA-Interim reanalysis. A relaxation zone of eight grid points was used at the lateral boundaries of the domain (Davies, 1976). The zonal and meridional wind components, atmospheric temperature, specific humidity, surface pressure and soil moisture and soil temperature were updated every 6 model hrs as lateral boundary conditions and interpolated to hourly distributions. They were introduced as initial conditions across the domain.

The IWV in ALARO-SURFEX has been derived using the model pressure, the surface pressure and the specific humidity at each model level. Since the GNSS station height can differ from the height of the surface layer in the model, an altitude correction was applied. First, the model surface layer height at the GNSS station was determined by a bilinear interpolation from the surrounding four grid boxes with the centre coordinates at the neighbouring grid box in the model to the GNSS station




coordinates. Next, from this height difference, the pressure at the GNSS station was calculated using the barometric formula (Oceanic and Administration, 1976):

$$P = P_{\text{sfcm}} \frac{T2m}{T2m + L_{\text{b}} \left(h_{\text{stat}} - h_{\text{m}}\right)}^{\frac{(g_0 \ M)}{(R^* \ L_{\text{b}})}} \tag{6}$$

where $P_{sfcm}$ is the model surface pressure of the nearest grid box to the GNSS station, $T2m$ is the model 2 m temperature of the nearest grid box to the GNSS station, $L_b$ is the standard lapse rate of -6.5 K km$^{-1}$, $h_{stat}$ is the height of the GNSS station, $h_m$ is the horizontally interpolated model height of the grid boxes nearest to the GNSS station, $g_0$ is the gravitational acceleration (9.80665 m s$^{-2}$), $M$ is the molar mass of the Earth's air (0.0289644 kg mol$^{-1}$) and $R^*$ the universal gas constant (8.3144598 J mol$^{-1}$ K$^{-1}$).

The IWV has been calculated by vertically integrating the water vapour content, starting from the pressure at the GNSS station up to the pressure corresponding with a height of about 20 km. This height of 20 km is well above the tropopause so with this upper limit, we capture the entire columnar water vapour amount.

## 2.4 Methodology

From the entire reprocessed GNSS dataset, we selected stations based on the following criteria: (i) the station had to fit within the model domain, (ii) the data length of the station had to cover at least 10 yrs, and (iii) the selected months from the dataset of the station had to cover at least 15 days with non-missing values. These criteria resulted in a total of 100 GNSS stations, for the 19-yr period of 1996-2014 (Fig. 1).

For the analysis, we calculated the IWV time series from the four neighbouring grid boxes nearest to the station coordinates from both ERA-Interim and ALARO-SURFEX. For this purpose, both reanalysed and modelled IWV were archived at 6-hr time steps (00 UTC, 06 UTC, 12 UTC, 18 UTC) starting at 1 January 1996. For subsections 3.1,3.2 and 3.3, we calculated the monthly means from these IWV values at the different stations. In this process, we identified one specific GNSS station Saint Jean des Vignes in France (SJDV, http://epncb.oma.be/_networkdata/siteinfo4onestation.php?station=SJDV00FRA) that showed large differences with both ALARO-SURFEX and ERA-Interim. These differences appear in the same order of magnitude for both models and are constant in time. They do not seem to be related to orographic difference between the model grid boxes and the location of the GNSS station, as the difference in altitude between the model height and the GNSS station is 184 m. Therefore, the differences could be related to the GNSS station itself and was removed from the dataset.

From the monthly mean values, we computed different types of variability: the inter-annual variability and the intra-annual variability. In both cases we use the average IWV values over all stations for each year i and each month j, with i=1..N and j=1..M, and N=19 and M=12. The inter-annual variability is associated with the year-to-year variability in IWV and is computed in two steps: (i) compute the monthly IWV anomalies w.r.t. the monthly mean over the 19 years; (ii) calculate the standard deviation of these anomalies for each month (resulting in 12 monthly values). The intra-annual variability refers to variability at the time scale of less than 1 year and is computed in two steps as well: (i) take the yearly IWV anomalies w.r.t.



5  the yearly mean values over the 12 months and (ii) calculate the yearly standard deviation of these anomalies (resulting in 19
yearly values).

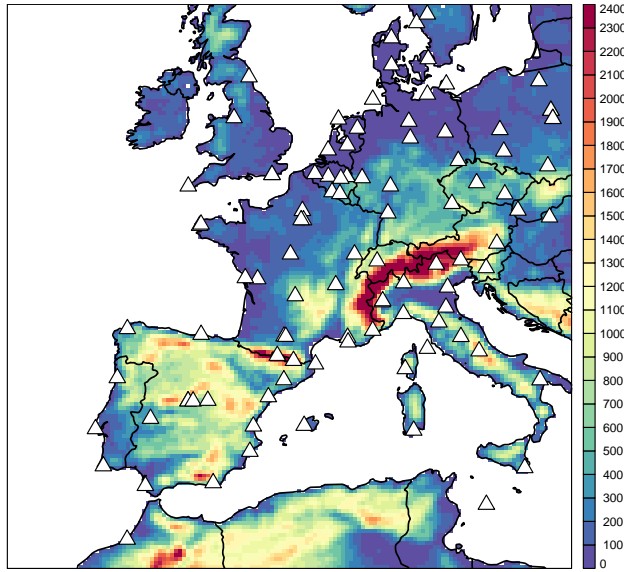

**Figure 1.** The domain used within this study at 20 km horizontal resolution centred at 46.47°N and 2.58 ° E with a dimension of 149x149
horizontal grid points. The distribution of the Global Navigation Satellite Systems (GNSS) stations that fulfil the criteria for the validation of
the models. The background colour gradient represents the orography of the domain.

The significance of the differences between ERA-Interim/ALARO-SURFEX and the GNSS observations was assessed by
applying the Kolmogorov-Smirnov (K-S) test. This test determines whether the datasets have the same (continuous) distribu-
tion. As integrated water vapour follows a non-normal distribution (Foster et al., 2006), the advantage of using this test is that

10  it does not make any assumption on the distribution of the data and is non-parametric (von Storch and Zwiers, 1999). The
significance was tested for the monthly values over the entire simulated 19-yr period at the 5% significance level.

## 3  Results

### 3.1  Dataset comparison

The IWV monthly values (all stations together) show a correlation coefficient of 0.99 and 0.98 for ERA-Interim and ALARO-
SURFEX respectively with the IWV derived from GNSS (Fig. 2). The spread of the model versus GNSS IWV values increases
with increasing IWV values. The slopes of the linear regression lines of both the ERA-Interim and ALARO-SURFEX IWV
values with respect to the corresponding GNSS IWV values are close to 1, resp. being 0.997 and 0.962. The lower slope for the
ALARO-SURFEX vs. GNSS linear regression is due to overestimation of ALARO-SURFEX w.r.t. the GNSS IWV (positive




5   IWV bias) at the low IWV range, and an increasing IWV underestimation (increasing negative biases) in the middle and upper
range of IWV Values (see Table 1).

The linear regression slopes of ERA-Interim IWV vs. GNSS IWV are increasing for the different IWV ranges considered in
Table 1), and is larger than 1 for the upper range (above 25 kg m$^{-2}$). This is due to an increasing number of ERA-Interim IWV
outliers above the 1:1 line with increasing IWV values. For all IWV ranges, ERA-Interim has a positive IWV bias w.r.t. GNSS,

10  with similar values. The linear regression slopes of ALARO-SURFEX IWV vs. GNSS IWV are lower than 1 for all IWV ranges
(Table 1). Despite smaller (but negative) biases by ALARO-SURFEX for all IWV ranges, the standard deviations are larger for
both middle and upper range (below 10 kg m$^{-2}$ and 10-25 kg m$^{-2}$). This indicates a larger variability of ALARO-SURFEX w.r.t.
GNSS around the 1:1 line than the ERA-Interim w.r.t. GNSS. This is due to the lack of assimilated ground-based observations
by ALARO-SURFEX (Ning et al., 2013).

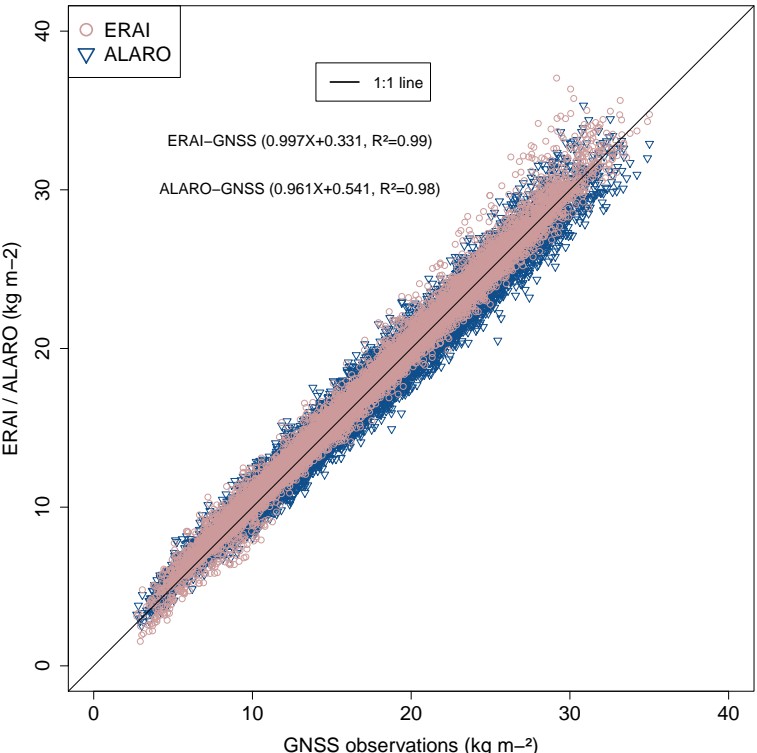

**Figure 2.** The scatterplot of the monthly Integrated Water Vapour (IWV, kg m$^{-2}$) values for all station locations for the 19-yr period of
1996-2014 of ERA-Interim and ALARO-SURFEX with respect to the Global Navigation Satellite Systems (GNSS) observations. The black
line represents the 1:1 fitted line with intercept 0 and slope 1.



**Table 1.** Statistics related to the scatterplot of the monthly Integrated Water vapour (IWV, kg m$^{-2}$) values.

| | | All ranges | Lower than 10 kg m$^{-2}$ | Between 10 and 25 kg m$^{-2}$ | Above 25 kg m$^{-2}$ |
|---|---|---|---|---|---|
| ERA-Interim | Slope | 0.997 | 0.951 | 0.988 | 1.013 |
| | Bias | 0.276 | 0.301 | 0.239 | 0.365 |
| | Std | 0.618 | 0.557 | 0.515 | 1.014 |
| ALARO-SURFEX | Slope | 0.962 | 0.859 | 0.952 | 0.845 |
| | Bias | -0.091 | 0.184 | -0.124 | -0.252 |
| | Std | 0.842 | 0.566 | 0.792 | 1.172 |

## 3.2 Yearly and seasonal variability

The mean GNSS-based IWV over the 19-yr period is 16.31 kg m$^{-2}$ and is higher for ERA-Interim with 16.58 kg m$^{-2}$ and lower with ALARO-SURFEX with 16.22 kg m$^{-2}$ (Fig. 3a). Both ERA-Interim and ALARO-SURFEX are able to reproduce the yearly IWV variability. For all data sets, the inter-annual variability is very small compared to the intra-annual variability (Fig. 3a, Table 2). The mean inter-annual variability is 1.20, 1.17 and 1.16 kg m$^{-2}$ for GNSS, ERA-Interim and ALARO-SURFEX respectively, whereas the mean intra-annual variability is 5.57, 5.52 and 5.39 kg m$^{-2}$ for the corresponding datasets. The IWV overestimation of ERA-Interim w.r.t. GNSS is a persistent feature for all the years, whereas ALARO-SURFEX has a positive IWV bias w.r.t. GNSS at the beginning of the year and a negative IWV bias in the summer periods (Fig. 3b). As a result, the ALARO-GNSS IWV difference time series show a strong seasonal behaviour. This results in a mean IWV difference of -0.08 kg m$^{-2}$ by ALARO-SURFEX and 0.28 kg m$^{-2}$ by ERA-Interim w.r.t. GNSS.



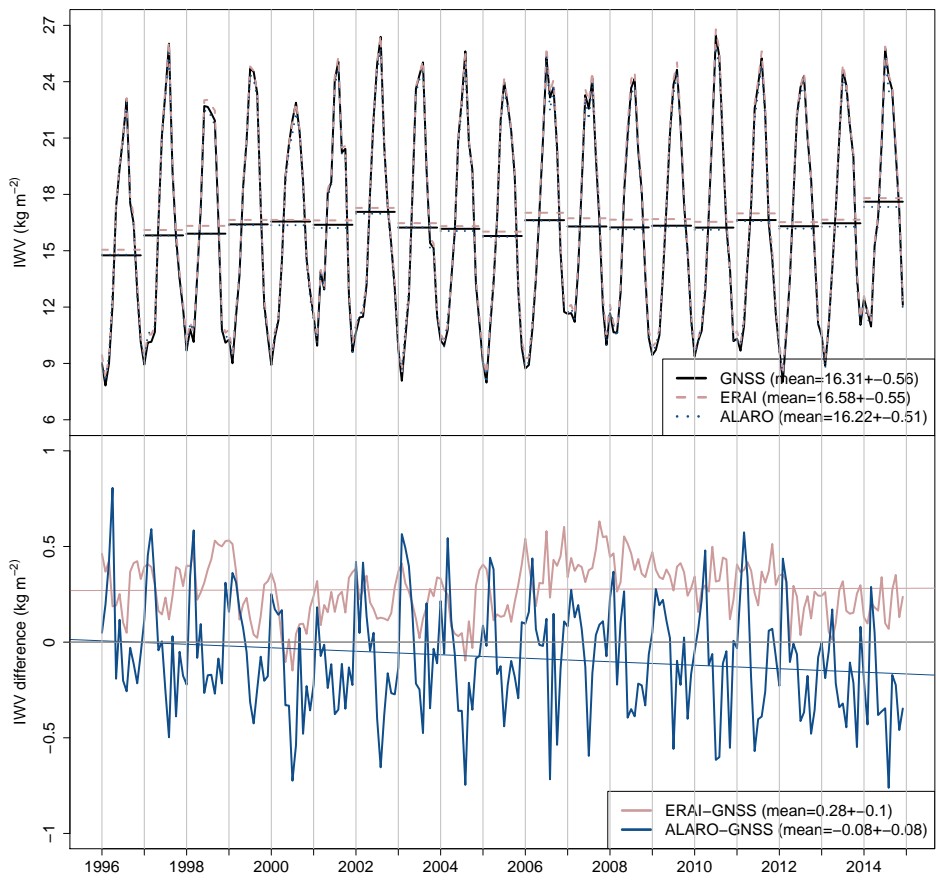

**Figure 3.** The inter-annual variability of monthly averaged Integrated Water Vapour (IWV, kg m$^{-2}$) from 1996 to 2014 for (top) the Global Navigation Satellite Systems (GNSS) observations, the ERA-Interim and the ALARO-SURFEX simulations and (bottom) absolute differences between ERA-Interim and ALARO-SURFEX and the observations. The horizontal lines in (top) represent yearly means and the mean value and standard deviation over all years is quoted in the inset panel. The horizontal lines in (bottom) represent the zero line (in black) and the linear fit through both datasets (ERA-Interim vs. GNSS in orange and ALARO-SURFEX vs. GNSS in blue) .

**Table 2.** Statistics related to the monthly Integrated Water Vapour (IWV, kg m$^{-2}$) values.

| Technique | Mean IWV over 19-yr (kg m$^{-2}$) | Inter-annual | | Intra-annual | |
|---|---|---|---|---|---|
| | | Mean (kg m$^{-2}$) | Std (kg m$^{-2}$) | Mean (kg m$^{-2}$) | Std (kg m$^{-2}$) |
| GNSS | 16.31 | 1.20 | 0.27 | 5.57 | 0.31 |
| ERA-Interim | 16.58 | 1.17 | 0.26 | 5.52 | 0.32 |
| ALARO-SURFEX | 16.22 | 1.16 | 0.26 | 5.39 | 0.31 |



All seasons during the 19-yr period present an overestimated IWV by ERA-Interim, except for two summer seasons in 2000 and 2004. These two summers correspond to anomalously low temperatures in ERA-Interim. The seasonality of the ERA-Interim vs. GNSS IWV differences is larger for the first years of the time series and is less pronounced afterwards. A possible explanation is the increased data assimilation in ERA-Interim in the more recent years. On the other hand, the ERA-Interim IWV biases w.r.t. GNSS seem to be rather constant in time. This is in contrast to the ALARO-GNSS IWV biases, which seem to decrease in time (drifting) (Fig. 3b). The seasonal cycle of ALARO-SURFEX exists for all the years, with a peak of overestimated IWV in spring.

The overall IWV seasonal cycle, averaged over all the GNSS station locations, is shown for the three datasets in Fig. 4a. The highest value is obtained in August (about 25 kg m$^{-2}$) and the lowest in February (about 10 kg m$^{-2}$). The standard deviations are larger in summer for all datasets. ERA-Interim overestimates the IWV values for all months with a fairly constant bias, on average 0.28 kg m$^{-2}$. They give significant differences in winter and spring to the GNSS IWVs (Fig. 4b) because of the lower IWV values in these seasons.

Meanwhile, the ALARO-SURFEX IWV biases with respect to GNSS IWVs are more variable throughout the year. ALARO-SURFEX is relatively good in simulating the IWV in winter, autumn and spring with 0.03 kg m$^{-2}$, -0.18 kg m$^{-2}$ and 0.13 kg m$^{-2}$ respectively, but significantly underestimates IWV in summer with an average of -0.34 kg m$^{-2}$ (Fig. 4b). This results in a month-to-month standard deviation by ALARO-GNSS of 0.84 kg m$^{-2}$, whereas ERAI-GNSS only presents a standard deviation of 0.62 kg m$^{-2}$ (Fig. 4b). Furthermore, the monthly standard deviations that represent the year-to-year variability and station-to-station variability are on the average 0.6 kg m$^{-2}$ for ERA-GNSS and 0.8 kg m$^{-2}$ for ALARO-GNSS, resulting in a 35% larger variability of ALARO-GNSS.



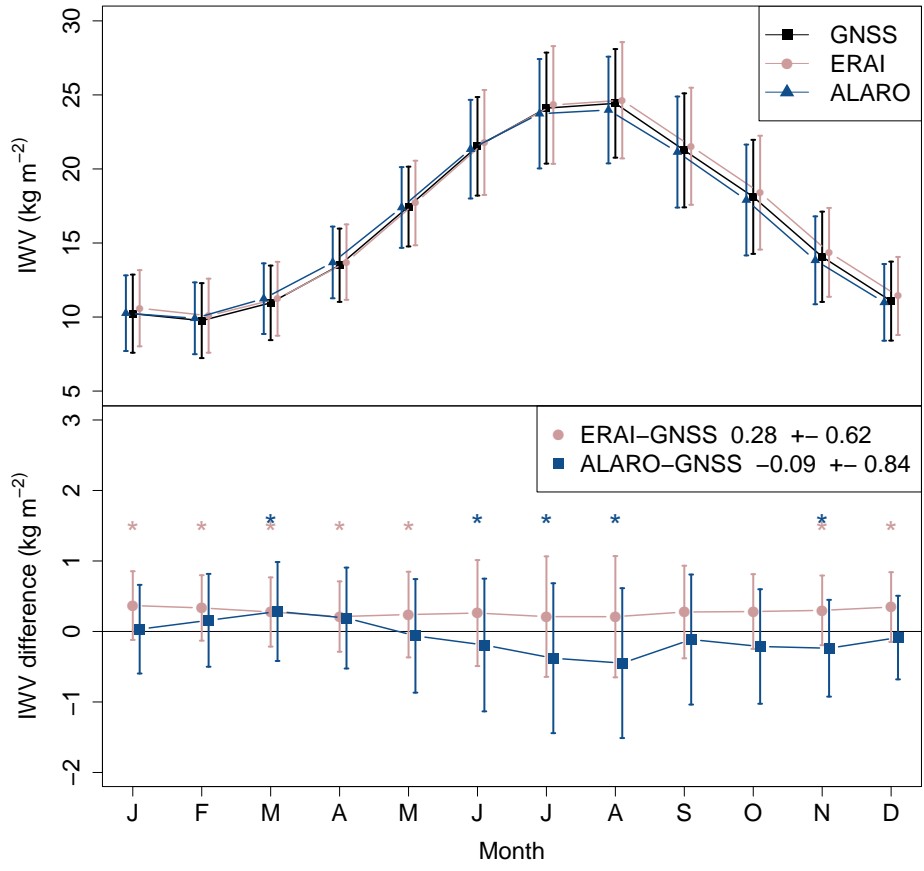

**Figure 4.** Monthly averaged Integrated Water Vapour (IWV, kg m$^{-2}$) (top) from Global Navigation Satellite Systems (GNSS) and modelled by ERA-Interim and ALARO-SURFEX and (bottom) differences between ERA-Interim and the GNSS observations and between ALARO-SURFEX and GNSS observations averaged over the 19-yr period of 1996-2014 and all GNSS stations. The vertical bars represent the standard deviations for the 19 years and 100 stations. The * represent the statistically significant IWV differences using the K-S test.

The seasonal variability visible in the IWV difference between ALARO-SURFEX and GNSS can be related to the seasonal variability of the temperature and the precipitation bias present in the model. The average precipitation over all grid boxes nearest to the GNSS stations shows a wet bias of 5-38% (Fig. 5a) compared to the 0.22° ECA&D E-OBS dataset (Haylock et al., 2008). E-OBS is a daily high-resolution gridded observational dataset, which consists of the daily mean temperature and the daily accumulated precipitation. The most recent version v14.0 was selected on the 0.22° rotated pole grid, corresponding to a 25 km horizontal resolution in Europe.

The overall wet bias is associated with an overall cold bias of -0.5°C to -1.8°C when averaging the 2 m temperature mea-
surements over the grid boxes that contain the GNSS station locations (Fig. 5b). These findings are in agreement with the general model performance of ALARO-SURFEX for the 10-yr validation period of 1991 to 2000 (Berckmans et al., 2017).





Both precipitation and temperature follow a clear seasonal cycle with strong biases in winter that increase in early spring and subsequently decrease in summer, when they are minimal, followed by gradually increasing biases again in autumn. The seasonal cycle of precipitation bias coincides well with the seasonal cycle of IWV bias for autumn, winter and spring (Fig. 4b).

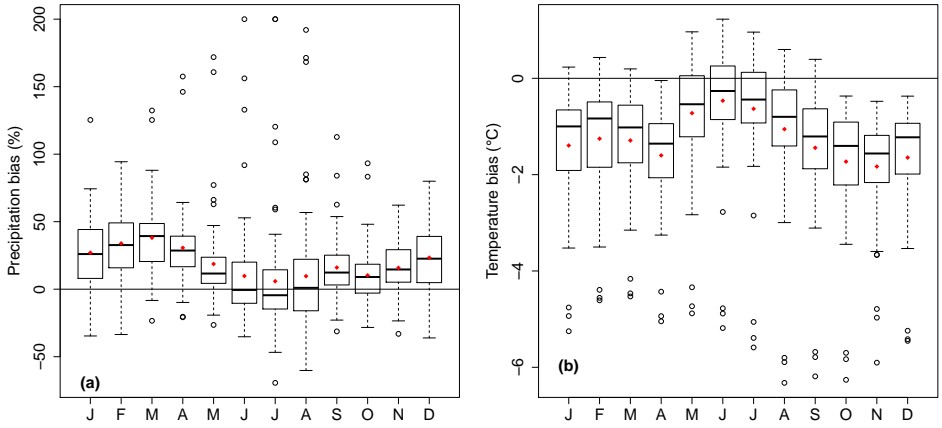

**Figure 5.** Monthly bias (presented by boxplots) by ALARO-SURFEX averaged over the 19-yr period of 1996-2014 and all grid boxes nearest to the Global Navigation Satellite Systems (GNSS) stations of (a) daily accumulated precipitation (%) and (b) daily mean 2 m temperature (°C) with respect to E-OBS. The boxplots indicate the standard deviation above and below the mean of the data, the horizontal bars represent the median of the data, the red diamonds represent the mean of the data, and the outliers are shown by open circles.

The largest difference between the ALARO-SURFEX IWVs and the GNSS-based IWVs occur in summer (Fig. 4b). On the contrary, the smallest temperature and precipitation bias appears in summer. This can be explained by a feedback between the land surface and the atmosphere that is strongest during summer (Seneviratne et al., 2010). Even though the temperature biases are smallest in summer, the means are still negative with values of -0.45 °C, -0.63 °C and -1.06 °C for June, July and August respectively. The lower temperatures in summer as compared to the observations are likely to lead to lower evapotranspiration rates. This result is in agreement with the findings by Ning et al. (2013).

The water vapour-evapotranspiration feedback is less dominant in the winter, when large-scale stratiform systems determine the weather and less interaction exists between the land surface and the atmosphere (Koster et al., 2000). Moreover, the precipitation is largely overestimated in winter (Fig. 5a) which corresponds to a larger overestimation of IWV (Fig. 4b). During autumn and spring, the previously explained land surface-atmosphere interaction exists for a selected number of stations. More specifically, it exists for 6 stations in autumn and 4 stations in spring (Fig. 5, Fig. 4b), as the region covering our stations hosts different climate regimes. Therefore, the coupling between the temperature and the IWV depends on the coupling strength between the land surface and the atmosphere.



### 3.3 Spatial variability

The IWV differences are very different among the individual sites and vary from -1.4 kg m$^{-2}$ to +4.6 kg m$^{-2}$ in winter (Fig. 6a,b) and from -2.0 kg m$^{-2}$ to values above +2 kg m$^{-2}$ in summer (Fig. 6c,d). From these figures, we could not note any latitudinal or longitudinal dependence of the IWV biases between ERA-Interim or ALARO-SURFEX, and GNSS. We will therefore concentrate on the dependence of the IWV biases on the altitude of the GNSS sites. As we found a clear seasonal variation of the ALARO-GNSS IWV biases in the previous section, we will discriminate between winter (DJF) and summer (JJA) in the discussion here.

ERA-Interim overestimates the IWV in winter for 91 stations, of which 71 stations show a difference less than 0.5 kg m$^{-2}$ (Fig. 6a), resulting in a mean difference of 0.34 kg m$^{-2}$. The IWV differences are similar for stations located in flat areas (< 100 m), at middle altitudes (> 100 m & < 1000 m), and at high altitudes (> 1000 m) with values of 0.34 kg m$^{-2}$, 0.36 kg m$^{-2}$ and 0.25 kg m$^{-2}$ respectively (Table 3). Besides, the standard deviations are very similar for the different altitude ranges.

However, at some locations in southern France (Aix-en-Provence), at the southern coastline of Spain (Almeria) and north-eastern Italy (Bolzano), ERA-Interim shows very high positive IWV differences with GNSS of more than 1.5 kg m$^{-2}$ (Fig. 6a). One of these stations, Bolzano, is located at 278 m and the height in ERA-Interim is 1579 m. For all stations with height differences of more than 500 m between the GNSS station and the ERA-Interim orography, the mean IWV difference is 0.76 kg m$^{-2}$ (Table 3). For GNSS stations with such high topographic differences that are located in mountainous regions with large topographic gradients, the spatial representativeness of the IWV field above the GNSS station by the four surrounding ERA-Interim grid points can be questioned. In these cases, the correction of the ERA-Interim IWV value for the height difference between the ERA-Interim surface grid points and the GNSS antenna is insufficient.



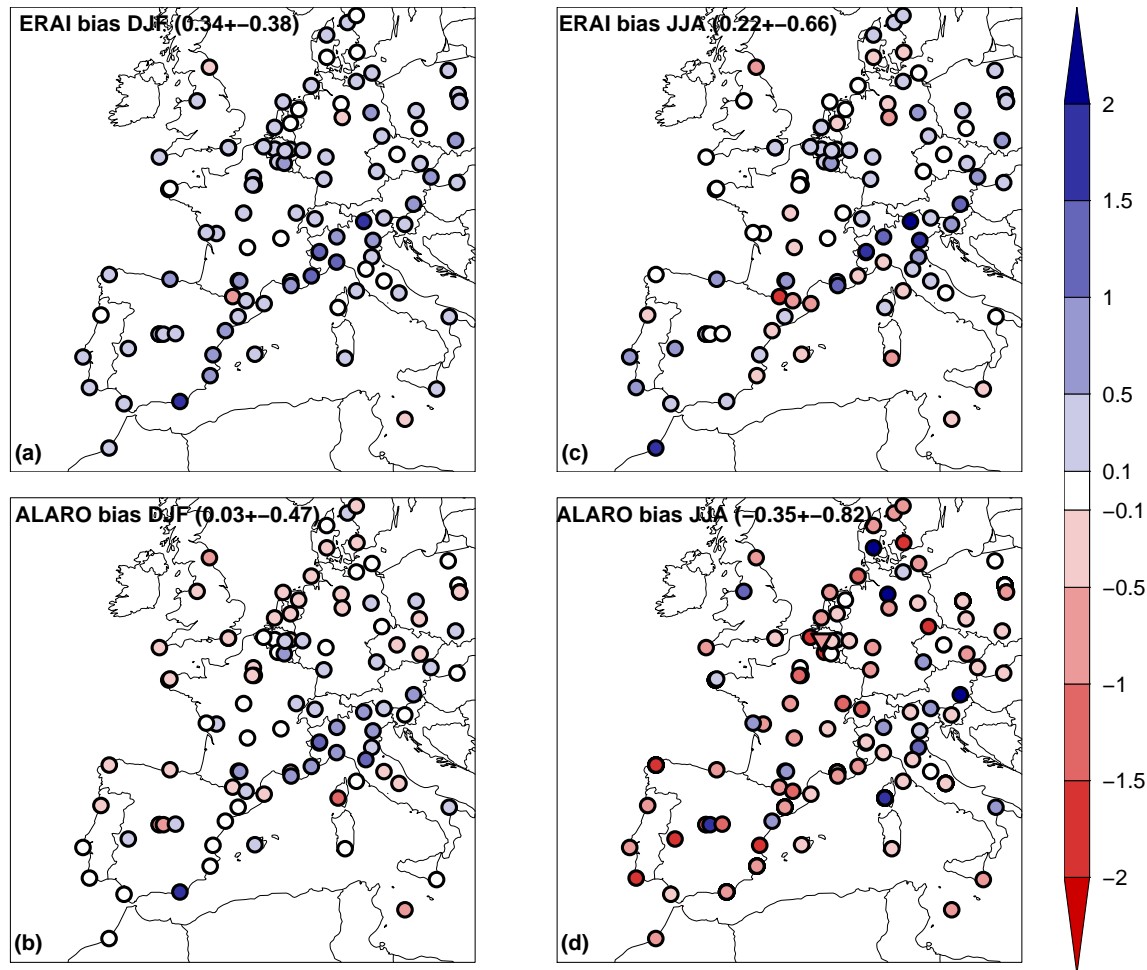

**Figure 6.** The spatial distribution of the Integrated Water Vapour (IWV, kg m⁻²) of the difference between (a,c) ERA-Interim and the Global Navigation Satellite Systems (GNSS) observations and between (b,d) ALARO-SURFEX and the GNSS observations for (a,b) winter (DJF: December-January-February) and (c,d) summer (JJA: January-February-March), averaged over the 19-yr period of 1996-2014. Significance of the differences was tested between the ERA-Interim/ALARO-SURFEX and observations for each station. No significant differences are found at the 5% significance level.



**Table 3.** Classification of the IWV differences based on altitude characteristics. The values between brackets correspond to the standard deviations.

|  | Number stations | Winter Bias | | Summer Bias | |
| --- | --- | --- | --- | --- | --- |
|  |  | ERA-Interim | ALARO-SURFEX | ERA-Interim | ALARO-SURFEX |
| All stations | 100 | 0.34 (0.29) | 0.03 (0.44) | 0.22 (0.47) | -0.35 (0.62) |
| Stations < 100 m | 44 | 0.34 (0.29) | -0.04 (0.45) | 0.16 (0.46) | -0.69 (0.64) |
| Stations > 100 m and < 1000 m | 52 | 0.36 (0.30) | 0.07 (0.44) | 0.34 (0.47) | -0.18 (0.61) |
| Stations > 1000 m | 4 | 0.25 (0.25) | 0.34 (0.32) | -0.60 (0.51) | 1.28 (0.45) |
| Stations with altitude difference > 500 m | 9 (ERAI) \| 3 (model) | 0.76 (0.37) | 0.27 (0.53) | 0.87 (0.64) | 1.71 (0.72) |
| Stations with altitude difference < 500 m | 91 (ERAI) \| 97 (model) | 0.30 (0.28) | 0.02 (0.43) | 0.15 (0.45) | -0.41 (0.61) |

In winter, a large number of stations (about 80%) show smaller differences between ALARO-SURFEX and GNSS than between ERA-Interim and GNSS (Fig. 6b). In total, 81 stations present IWV differences between -0.5 m$^{-2}$ and +0.5 m$^{-2}$. This results in a mean difference of only 0.03 kg m$^{-2}$ compared to 0.34 kg m$^{-2}$ by using ERA-Interim. The mean difference is smallest for stations in flat areas (< 100 m) with -0.04 kg m$^{-2}$, and increases for stations at higher altitudes (> 100 m & < 1000 m) with 0.07 kg m$^{-2}$. At highest altitudes (> 1000 m) the IWV difference is much larger with 0.34 kg m$^{-2}$ (Table 3). Therefore, ALARO-SURFEX is more sensitive than ERA-Interim to the station height for modelling the appropriate IWV. However, the variability (indicated by the standard deviations) is similar for the different altitudes.

Similarly as for the winter, ERA-Interim overestimates IWV for 69% of the GNSS stations in summer (Fig. 6c). This results in an average value of 0.22 kg m$^{-2}$ which is lower than the winter averaged value by ERA-Interim. Furthermore, 73 stations give small differences of -0.5 kg m$^{-2}$ to +0.5 kg m$^{-2}$ between ERA-Interim and the observations. Smallest IWV differences occur for the GNSS stations located in flat areas with an average of 0.16 kg m$^{-2}$ (< 100 m) (Table 3). For stations at higher altitudes (> 100 m & < 1000 m) the mean IWV difference increases to 0.34 kg m$^{-2}$ and is -0.6 kg m$^{-2}$ for stations at altitudes

above 1000 m (Table 3). This deterioration of the IWV difference with highest altitudes of the GNSS stations could not be distinguished for winter.

In contrast to the general overestimation of ERA-Interim, ALARO-SURFEX underestimates IWV in summer for 76% of the stations (Fig. 6d). The geographical distribution of the ALARO-SURFEX biases with GNSS is different in summer and winter. Only 39 stations present small IWV differences between -0.5 kg m$^{-2}$ and +0.5 kg m$^{-2}$ when simulated by ALARO-SURFEX.

Large negative IWV differences occur for GNSS stations located in flat areas (< 100 m) with an average of -0.69 kg m$^{-2}$ and large positive IWV differences exist for GNSS stations located at high altitudes (> 1000 m) with an average of 1.28 kg m$^{-2}$ (Table 3). The IWV is best represented by the stations between 100 m and 1000 m of altitude.

In summary, we find that the dependence of the model-GNSS IWV bias on the altitude of the GNSS station is the strongest in summer and also the strongest in ALARO-SURFEX. The IWV biases increase with increasing altitude for the stations,

both for summer and winter, except in winter with ERA-Interim. For stations with height differences larger than 500 m, IWV differences from 0.75 kg m$^{-2}$ up to 1.71 kg m$^{-2}$ are observed except for ALARO-SURFEX in winter.



## 3.4 Hourly variability

The IWV varies during the day due to changes in the solar radiation. Therefore, it is of interest to investigate the capability of the model to capture this daily variations. The IWV is observed on a high temporal resolution by the GNSS stations, hence it is a valuable technique for validating the ability of regional climate models to represent the diurnal cycle (Ning et al., 2013; Wang et al., 2007). It is not possible to investigate the exact comparison of the amplitude of the diurnal cycle, as a phase shift could be present (Ning et al., 2013) and because this shift is not detectable as we only archived ERA-Interim data every 6 hrs. So, hereafter we investigate the seasonal variations of the diurnal cycle.

5       The observed IWV diurnal cycle is the strongest in summer with an amplitude of approx. 1 kg m$^{-2}$ and the weakest in winter with an amplitude of less than 0.5 kg m$^{-2}$ (Fig. 7). For all seasons, the observed IWV peaks in the evening at 18 UTC and is lowest at 06 UTC (Fig. 7). These characteristics of the IWV diurnal cycle are captured similarly by the three datasets. Different factors contribute to this diurnal variation, the most important factor being the temperature increase during the day that drives the evaporation due to increased water vapour holding capacity, which increases the IWV. During the night, condensation takes
10     place, which causes cooling and a decrease of the IWV (Ortiz de Galisteo et al., 2011). The diurnal variability is controlled by varying factors such as precipitation and wind speed (Wang et al., 2007; Wang and Zhang, 2009).

        The IWV diurnal differences between ERA-Interim or ALARO-SURFEX and the GNSS-based IWVs are examined (Fig. 8), again for the different seasons separately. ERA-Interim overestimates IWV for all seasons and for all hours, except for 06 UTC and 12 UTC in summer (Fig. 8a). During winter, ERA-Interim presents largest differences of 3-4% that are consistent
15     from midnight till evening. For spring, summer and autumn, the differences are largest at 00 and 18 UTC (around 2 to 3%), and smallest at 06 and 12 UTC (+/- 1%).





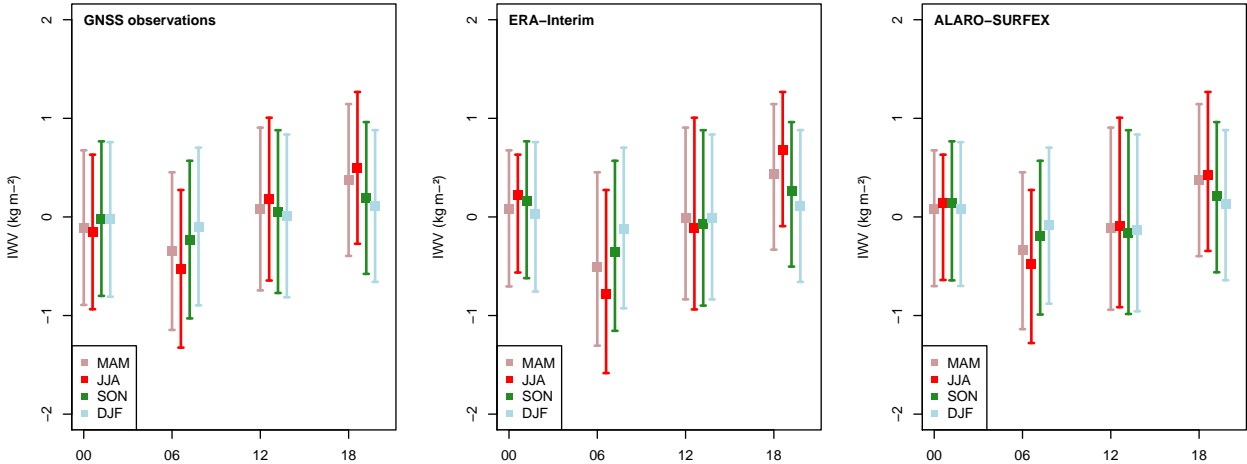

**Figure 7.** Amplitude of the diurnal cycles of the Integrated Water Vapour (IWV) (kg m$^{-2}$) for (left) GNSS observations, (middle) ERA-Interim and (right) ALARO-SURFEX subtracted from the mean seasonal cycle and averaged over the 19-yr period of 1996-2014 and all GNSS stations and their neighbouring grid boxes in the models for spring (MAM: March-April-May), summer (JJA: June-July-August), autumn (SON: September-October-November) and winter (DJF: December-January-February). The error bars represent the standard deviations for the year-to-year variability.





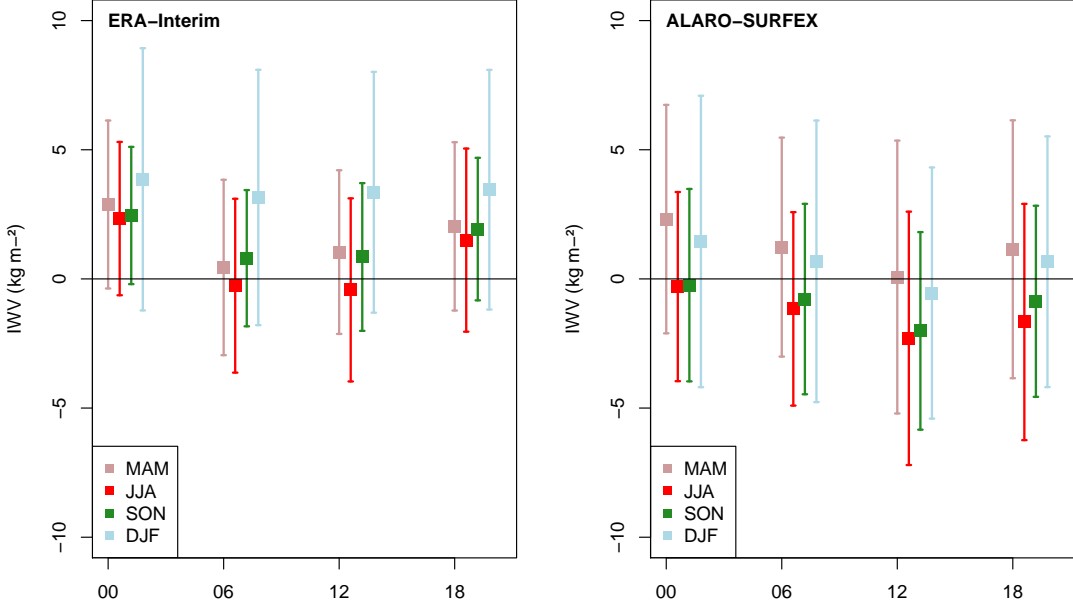

**Figure 8.** Amplitude of the diurnal cycles of the Integrated Water Vapour (IWV) relative difference (%) between (left) ERA-Interim and (right) ALARO-SURFEX with the GNSS-derived IWVs averaged over the 19-yr period of 1996-2014 and all GNSS stations and their neighbouring grid boxes in the models for spring (MAM: March-April-May), summer (JJA: June-July-August), autumn (SON: September-October-November) and winter (DJF: December-January-February). The error bars represent the standard deviations.

In contrast to ERA-Interim, the representation of the IWV diurnal variations by ALARO-SURFEX is now highly different between spring and winter on the one hand and summer and autumn on the other hand (Fig. 8b). During summer and autumn, the IWV is underestimated by the model at all time, with increasing differences from midnight (when they are close to zero) towards noon and decreasing again in the afternoon. We believe that the strong underestimation at 12 UTC is due to the mechanism of an evaporation-temperature interaction that is most pronounced during the day due to incoming solar radiation. The same diurnal variations of the ALARO-GNSS biases are found for spring and winter, but are offset with 1 to 2%, so that the best agreement is now found for the observations at noon, and the worst at midnight. So ALARO-SURFEX simulates IWV

5    better at 00 UTC for summer and autumn at 12 UTC for spring and winter (Fig. 8b).

In contrast to summer, ALARO-SURFEX is very good in representing the IWV daily cycle in winter and outperforms ERA-Interim (Fig. 8a,b). Moreover, it performs well for spring with similar values than ERA-Interim. A possible explanation might be that the water vapour during winter and spring is more controlled by large-scale stratiform systems, that are forced by ERA-Interim. Therefore, the behaviour of ERA-Interim and ALARO-SURFEX in winter and spring is more comparable than

10    in summer and autumn. Moreover, it seems that ALARO-SURFEX further improves the representation of the IWV daily cycle for these seasons because of the upgraded microphysics scheme.



## 4   Conclusions

This study explored the potential of the ERA-Interim reanalysis and the Regional Climate Model (RCM) ALARO-SURFEX in simulating the atmospheric water vapour content derived from ground-based GNSS observations. The reprocessing efforts made within EUREF during the course of the the COST Action ES1206 (GNSS4SWEC) led to the development of an extremely valuable dataset for climate monitoring. Within our model domain, 100 stations were selected from the EPN-Repro2 dataset covering the 19-yr period 1996-2014. The IWVs from ERA-Interim and ALARO-SURFEX have been validated by IWVs derived from GNSS.

Over the complete period and over all stations, the correlation between the IWVs derived from GNSS and the IWVs modelled by both ERA-Interim and ALARO-SURFEX were 0.99 and 0.98 respectively. The variability of IWV from ERA-Interim was lower than from ALARO-SURFEX, due to the assimilated observations in ERA-Interim. The distributions of the IWV determined from ERA-Interim and ALARO-SURFEX were closest to the GNSS-based values in the middle IWV range, i.e. between 10 and 25 kg m$^{-2}$. In the lower IWV range (i.e. < 10 kg m$^{-2}$) and in the upper IWV range (i.e. > 25 kg kg m$^{-2}$), ALARO-

SURFEX underestimated IWV with more than 10% w.r.t. GNSS whereas ERA-Interim slightly under -and overestimated IWV w.r.t. GNSS respectively.

For all datasets, the intra-annual variability was much higher than the inter-annual variability by a factor of 4 to 5. Both ERA-Interim and ALARO-SURFEX were capable of reproducing the observed yearly IWV cycle. However, ERA-Interim overestimated IWV for most years with an average of 0.27 kg m$^{-2}$, with a decreasing seasonality in the more recent years,

possibly due to the increased amount of data assimilated in ERA-Interim along the years. ALARO-SURFEX demonstrated a strong seasonal effect with overestimated IWV values in spring and underestimated IWV values for the summer periods.

The variability of the monthly differences was 25% higher by ALARO-SURFEX than by ERA-Interim, possibly due to the fact that no observations were assimilated by ALARO-SURFEX. The significant underestimation of IWV in summer by ALARO-SURFEX was related to the modelled precipitation and temperature bias. An overall cold and dry bias in the summer

led to lower evaporation rates and thus an underestimation of the IWV by ALARO-SURFEX. This mechanism was most pronounced in summer as land surface-atmosphere feedbacks are strongest in summer.

No clear latitudinal or longitudinal dependence of the IWV biases could be detected. Such effect is probably hampered by the seasonal variation of the biases. The dependence of the IWV on the altitude of the GNSS station was strongest in summer

and strongest for ALARO-SURFEX. The IWVs were corrected for a possible height difference between the surface model height and the actual height of the station. For stations with large height differences between the reanalysis or model grid box and the GNSS station, the IWV differences were highest of all stations.

The IWV peaks in the evening and reaches its minimum in the morning. The diurnal cycle has the largest amplitude in summer and was even enhanced by ERA-Interim. The diurnal cycle of ERA-Interim was most comparable to ALARO-SURFEX in

winter and spring. However, ALARO-SURFEX outperformed ERA-Interim in winter with smaller IWV differences. ALARO-SURFEX represented IWV best at midnight in summer and autumn and at noon in spring and winter.



This study focused on model validation at a fixed horizontal resolution of 20 km. At this resolution, the feedback between water vapour and other meteorological variables could lack a sufficient representation. Therefore, we suggest to validate the model at higher resolutions to study the feedback mechanism during the summer.

The reanalysis product of ERA-Interim has no assimilated GNSS observations. A clean comparison could be achieved by assimilating the GNSS observations in ALARO-SURFEX when downscaling the ERA-Interim data. Furthermore, the comparison of the reanalysis w.r.t. GNSS can be improved by investigating more in depth the daily and sub-daily cycle of IWV. Besides, a new reanalysis product is available called ERA5, that includes better assimilation, a higher horizontal resolution at 31 km and a better model than the original ERA-Interim product.

*Author contributions.*

Julie: simulation ALARO-SURFEX, analysis,text on analysis

Roeland: data ERA-Interim, ideas, help in the writing process

Eric: ideas, discussions, interpretation, writing process

Rosa: data EPNrepro2

Rafiq: providing lateral boundary conditions, providing the model, help in the writing process

*Acknowledgements.* This work has been supported by the COST Action ES1206 GNSS4SWEC (http://gnss4wec.knmi.nl). This research was funded by the Belgian Federal Science Policy Office under the BRAIN.be program as MASC contract no. BR/121/A2. The authors would like to thank Pieter De Meutter for providing scientific support in the model setup. E. Pottiaux and R. Van Malderen also wish to thank the Solar-Terrestrial Centre of Excellence (STCE) for its support in this study.





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
