# Peer review of "Validating the water vapour content from a reanalysis product and a regional climate model over Europe based on GNSS observations"

_Atmospheric Chemistry and Physics, 2018_

## Referee Comment (RC1) · Anonymous Referee #1 · 3 Jan 2019

This paper uses a new homogeneously reprocessed dataset of tropospheric delays (EPN Repro2) converted into Integrated water Vapor to evaluate the ability of one regional climate simulation performed with ALARO-0 coupled with SURFEX to simulate IWV. The methodology is first described and then results of comparisons between observations, ALARO and ERA-Interim at different time scales are presented and some interpretations are given. The main issue I have regarding this paper is that it concerns only one variable (except very short and speculative discussion on the link with temperature and precipitation bias) and one simulation and I feel that there is a lack of depth in the interpretation of the results which makes the relevance of this paper for the international community very reduced. Hence I invite the authors to explain

in more detail what the added value of this work is, and in particular, its context and perspectives, and to give an attempt at taking a Âń deep dive Âż in the results. There are a number of speculative conclusions that should not be included in the text. As is, it seems like the principal outcome of this project is to assert that very similar results than Ning et al. (2013) found are obtained (depiste a different model and a new re-processed dataset), and that the use of a RCM at 20km resolution does not have any added-value compared to the use of ERA-Interim.

More details (major and minor comments) are given below :

1. abstract : 'The model presents a cold and dry bias in summer that feedbacks to a lower evapotranspiration and results in too few water vapour.' => Speculative result : this analysis does not demonstrate the lower evapotranspiration. The correlation between spatial averaged seasonal cycle of temperature, precipitation and IWV does not explain anything.

2. Abstract : 'The spatial variability among the sites is high ' : in terms of what ? bias ? variability ?

3. Abstract : ' and shows a dependence on the altitude of the stations which is strongest in summer and by ALARO-SURFEX. ' what explains the strongest dependence in summer, and in ALARO-SURFEX ?

4. Introduction p2, l 6-20: several references to papers of this ACP special issue and/or which use GNSS observations to evaluate models/reanalyses are missing.

5. Introduction p2, l17 : ' Therefore, some authors suggested to be careful when validating model grid boxes with point observations (Ning et al., 2013). ' => Please explain in more detail what you did to address this point in your methodology. This is not obvious in the paper what is different in your methodology than in their. You can add a short sentence in the introduction following this sentence and include more details in section 2. Specifically, the fact that you compare two different grids (ERA-I at 75km

and ALARO at 20km) with point observations is not addressed in the current version.

6. Introduction : the challenges and motivations of getting a good representation of IWV in a RCM's simulation are not enough explicitly described and explained.

7. Section 2.1 : since ERA-Interim surface pressure and mean temperature are used to convert ZTD to IWV, wouldn't it be more faire to compare ZTD datasets instead of IWV ? This question may be beyond the scope of this study, however I am wondering how it helps ERA-Interim to get better results in the comparison ? Shouldn't be useful to compare Ps and Tm from ERA-I and ALARO with and without remapping of ALARO to ERAI grid ? Can this explain the stronger sensitivity of ALARO to altitude ?

8. Section 2.2, p4 l30 : ' ... the model topography using in ERA-Interim... ' => used.

9. Section 2 : I'm not sure to understand why the authors did not use the same methodology between ERA-Interim and ALARO to extract IWV at the GNSS stations. It seems that the height correction applied differs and it is not clear to me what is extracted at the closest grid point, what is bi-linearly interpolated. Please explain and add some numbers to have a way to compare the bias and standard deviation obtained in section 3 with the uncertainty resulting from these approximations/methodologies.

10. Section 2.4 : criteria iii): 15 days for a month is not very restrictive. How many months of less than 25 days do you have ? could you give an estimate of the impact the number of missing days can have on a monthly mean ? For instance, using ERA-I.

11. Section 2.4, criteria iii) : the data length of the station covers at least 10 years : does it mean that it contains at least 10 years with 12 months, or that there is a time lapse of 10 years between the first and the last measurements ? If the second interpretation is the good one, isn't it necessary to add another criteria with a minimum number of months contained in the time series ?

12. Section 2.4 : for the comparison with ERA-Interim and ALARO, it is not clear if the authors used exactly the same sampling than observations by removing simulated

values when observations are missing or not.

13. Section 3 : there is a problem with the line numbering throuhout this section, so I will not refer to line numbers when possible to avoid misunderstanding

a. P8, section 3.1, just before Fig.2: This sentence ' This is due to the lack of assimilated ground-based observations by ALARO-SURFEX (Ning et al., 2013) ' is speculative : lots of reasons can explain the differences between ALARO and ERA-Interim, not only the assimilation of ground-based observations. Dynamics differ, boundary-layer processes also, impact of vertical resolution, horizontal resolution, correction height methodology, the fact ERA-Interim is used to convert ATD into IWV etc. . ..

b. Figure 3a : it is very difficult to see ERAI and ALARO mean values : why don't you use solid lines also since colors differ ? Also, on Fig.3b, increase the width of the zero line.

c. P 9, Section 3.2, 6th and 7th line of this paragraph : please refer to Fig. 4 also.

d. P11, first paragraph : please indicate to the reader that you are analysing Fig.3b. Two aspects of Fig.3b would really need an attempt at explaining what happens, more than what is done in this paragraph (and because it will affect any attempt to compute a trend of IWV): 1) the break/jump in the ERA-Interim time series, with a seasonality which is larger for the first years of the time series until around 2005, with a drift of the bias between 1996 and 2005; and an increase of the bias between 2004 and 2006, with a reduced seasonality afterwards but another drift between 2006 and 2014 ; 2) the drift of ALARO (please modify your sentence since the bias of ALARO does not decrease in time, this is the absolute value of IWV which decreases increasing the negative bias.). If point 1) is due to a change of data assimilation, you have to check that and explain more precisely. At least, use some references.

e. P11, discussion on the seasonal cycle : At the end of the previous paragraph, the authors say 'The seasonal cycle of ALARO-SURFEX exists for all the years, with a

peak of overestimated IWV in spring.' and a few lines later they say ' ALAROSURFEX is relatively good in simulating the IWV in winter, autumn and spring with 0.03 kg m-2 , -0.18 kg m-2 and 0.13 kg m-2 respectively, but significantly underestimates IWV in summer with an average of -0.34 kg m-2 (Fig. 4b).' This seems a bit contradictory.

f. P11 : it could be interesting to compare the bias to the uncertainty due to correction height, conversion to IWV etc. . ..

g. P12 : I don't understand ' a wet bias of 5-38% '

h. P12 : please, justify why you can directly compare E-OBS and ALARO for precipitation. Also, concerning the sampling, did you use the same than for the comparison with GNSS (if you took into account missing values in your comparison between GNSS and ALARO, see my comment #12) ?

i. The link between the seasonal cycles of IWV, T and P shoudl be explained in more details, using references and other figures to justify the interpretations. And I think that you can not consider the averaged domain to explain the processes, you need to consider the spatial variability because of the very different processes involved in the arid southern part of the domain, western, central and scandinave parts. In particular, you explain the reduced IWV in summer by the lower temperature which leads to lower evapotranspiration. Another possible explanation, at least for southern stations, is that in spring more energy is converted into latent than into sensible heat fluxes, leading to a cold and wet bias, and in summer, the soil moisture is lacking due to an overconsumption in spring and consequently more energy is converted into sensible than in latent, decreasing the cold bias, and inducing a lack of moisture to reach the water-holding capacity of the atmosphere, generating a dry bias.

j. The section concerning spatial variability should be put and discussed before section 3.2, and at least before the seasonal cycle.

k. Please try to explain (see comment #3)

l. Section 3.4 : please explain the motivation : why is it important for a model to capture the diurnal cycle of IWV ?

m. Section 3.4, p19, last sentence : once agin, this sentence is very speculative !!

14. Discussion : what do you expect from a simulation at 20km resolution compared to ERA-Interim ? What are the added values of this new dataset ? of this methodology of comparison ? of this simulation etc. . . ? You conclude that at this resolution the feedback between water vapor and other variables could lack of a sufficient representation but at any moment you try to test this representation. What do you expect from an increased resolution ? And how will you check that feedbacks are better represented ? I would suggest to first focus on the reasons why the model is not perfect before trying to use higher resolution simulations that are very expensive in computational time and ressources. You have a lot of aspects that have not been explored in the physics of the model.

---

## Referee Comment (RC2) · Anonymous Referee #2 · 18 Mar 2019

The study is about a comparison of Integrated Water Vapour (IWV) between a product derived from ground based GNSS stations and two products from atmospheric models: the global ECMWF ERA-Interim reanalysis at 80 km resolution and a Regional Climate Model (RCM) ALARO-SURFEX at 20 km resolution. The comparison is undertaken over Europe for a 19-year period (1996-2014) thanks to an homogeneous reprocessed GNSS dataset "EPN Repro2" described in Pacione et al. (2017). Even though the authors conclude that both models reproduce reasonably well the behaviour of IWV from GNSS in terms of seasonal cycle and interannual variability, I found that the discrepancies noticed, in particular in terms of biases, lack of satisfactory explanations (they are most of the time highly speculative). The strengths and weaknesses of each model

are not exploited for an improved understanding of the different behaviours: the RCM has a high horizontal resolution (e.g. better descriptions of orography and sea-land contrast) but it is not constrained by observations ; ERA-Interim has a coarser resolution but the atmospheric state is constrained by observations (in particular in terms of water vapour). It is not clear at the end of the study to know if the RCM could be used for climate change scenarios with confidence. For example, could the trend in differences with GNSS noticed in Figure 3 over the 19-years jeopardize the confidence in trends that the RCM could simulate in a climate changing scenario ? I also have the impression that the added value of the "EPN Repro2" dataset does not show up since the conclusions from the study of Ning et al. (2013) with an older GNSS dataset over 11 years are rather similar to the ones reached by the authors for ERA-Interim. Since the domain is different (northern Europe) as well as the RCM some conclusions are necessarily different. From all these elements I am not favourable to the publication of this paper in Atmospheric Chemistry and Physics. To my opinion, the authors should go deeper in the analysis of the three datasets for an improved understanding of the strengths and weaknesses of their RCM in terms of IWV. They should ask themselves: what a reader of our paper can learn from the results and methodology we are presenting ? In its present form, most statements given in the paper are either descriptive or without proper explanations.

Please find below a number of specific comments (P = Page and L = Line ; from page 7 the line numbering does not make sense):

P1L3-5: You should also mention that you will also compare ERA-Interim analyses against IWV observations

P1L9-10: From this sentence the reader may think that this bias comes from the lateral boundary conditions. This is not obvious. When examining Figure 6, stations closer to the boundaries do not have biaises more consistent with ERA-Interim. I have the impression that there is more consistency with stations in the inner part of the domain (but this is probably just consistent problems between the two models with orography

over the Alps).

P1L11-12: This explanation is far from being convincing. Summer biases linked to evapotranspiration are generally producing warm/dry biaises or cold/wet biaises. A cold and dry bias is more likely to come from an underestimation of the downward radiation at the surface reducing both turbulent sensible and latent heat fluxes.

P1L20: I am surprized to learn that radiosonde and satellite instruments measuring water vapour are not adequate for the validation of climate models. I can understand that they are not sufficient but nevertheless there are useful climate records in terms of radiosonde measurements (e.g. the GRUAN programme) or satellite humidity from sounding and imaging instruments (e.g. AMSU-B or SSMI sensors ; Climate Monitoring Satellite Application Facility from EUMETSAT).

P1L4: I do not understand why the high temporal variability is important for climate applications, whereas I can understand that it is crucial for short-range weather forecasts.

P2L2: It is wrong to state implicitly that radiosondes cannot provide measurements in all weather conditions. They provide vertical profiles of atmospheric parameters in clear sky and cloudy conditions. The issue is more relevant for satellite measurements and particularly in the infra-red and the high frequency microwave spectral regions. Putting forward the argument for GNSS of high temporal and spatial resolutions is misleading. Radiosondes have a much better spatial resolution on the vertical than ZTD or STD (Slant Total Delay) values from GNSS receivers. On the other hand, geostationary satellites provide measurements every 15 min. This opposition of GNSS observations vs. "traditional systems" is presented is an unfair manner.

P2L6: GNSS signals are also delayed by the ionosphere. This delay is non negligible and has to be removed for a meaningful interpretation of the tropospheric part of the signal.

P2L9: The "water vapour weighted mean temperature" is a very specific concept in

order to convert the ZTD into a IWV. The proper reference is more likely Davis et al. (1985).

P3-4: Section 2.1 could be shorten. It is a very classical description (I found exactly the same with similar formulae in Ning et al., 2013). Since numerical models can easily compute ZTD values that are closer to the actual measurements, why are comparisons done in terms of IWV that require additional assumptions on the observation side ?

P4L12-17: What is the purpose of this discussion ? Are the numbers proposed for these 3 stations representative ? Could they be used to put an error bar on IWV from GNSS data (e.g. interpretation of Table 3) ? I am wondering how useful they can be given the strong seasonal cycle of IWV. Different values are expected between winter and summer seasons.

P5L25: I am wondering why the soil moisture and soil temperature are included in the lateral boundary conditions since soil processes only describe vertical transfers (no lateral transfers). It is rather surprising since the surface scheme SURFEX is very different from the ECMWF land surface model.

P5L8: A reference to the description of the AROME model is needed : Seity et al. (2011)

P5L27: Can you explain the sentence "they are introduced as initial conditions across the domain" since the previous sentence is about latereal boundary conditions ?

P6L1-8: It is not clear if this adjustment to orography is different from the one described for ERA-Interim (making reference to Hagemannn and Bengtsson (2003)) ? If it the case please explain why.

P6L11-12: The sentence can be simplified as "station up to a pressure level corresponding to a height of about 20 km (that is sufficient to capture the entire columnar water vapour)"

P6L17: replace "(Fig. 1)" by "displayed in Fig. 1"

P6L18: Statement already mentioned

P6L21-26: Six lines of explanations for one specific station are not necessary

P8L13-14: There is nothing to support such statement : "this is due to the lack of assimilated ground-based observations by ALARO-SURFEX". The paper of Ning et al. (2013) does not provide any support.

P9L17: The statement cannot be seen in Figure 3a

P9L23-2: Why the fact that there is strong seasonal cycle of the bias in ALARO-SURFEX explains the mean bias values ? The statement "This results in a mean IWV ..." does not make sense to me. It has more to do with positive and negative values between winter and summer.

P10: Figure 3 : add labels a) and b) as mentioned in the text. Once you have defined IWV and GNSS there is no need to rewrite again the meaning of the acronyms.

P10: Legend of Table 2 is incomplete

P11L4: Rewrite the sentence as : "explanation is the increased number of observations in the ERA-Interim data assimilation system in the most recent years"

P11L11-12: Rewrite the sentence as : "This bias in statistically significant in winter and spring because of lower IWV values in these seasons"

P13L3: The coincidence of the precipitation biases with the IWV biases for some seasons (autumn, winter, spring) is unclear to me (are you talking about the amplitude, the sign, ... ?).

P13L8: The link between the negative temperature bias in summer and lower evapo-transpiration rates is not straightforward and not explained.

P13L12: The link between positive bias in precipitation and in IWV is not straightforward. One could argue that if more water is lost by the atmosphere through precipitation the column contains less water vapour.

P14: The comment of Figure 6 is very limited. The biases along coastal regions that the ALARO-SURFEX model at higher resolution do not seem to handle better with respect to ERA-Interim are not discussed. Issues the orography over the Alps appear clearly but they are also presented and discussed in Table 3.

P16: Table 3: are the statistics for stations above 1000 m (only 4) reliable ?

P16L6: You should provide an explanation why ALARO-SURFEX is more sensitive to station height than ERA-Interim. This is not intuitive. Since this RCM has a higher resolution than the ECMWF model used for ERA-Interim, the differences should be smaller and therefore less sensitive to extrapolation assumptions.

P17: Section 3.4 on hourly variability is difficult to understand since as presented in Figure 7 and 8 the amplitude of the diurnal cycle varies during the day (different values at 00, 06, 12 and 18 UTC) with small end even negative values. This amplitude should be the difference between the minimum and maximum values over 24 hours. Moreover, I do not see the interest of presenting both Figures 7 and 8. To my opinion, Figure 8 is sufficient.

P19L4: The statement "we believe" should be avoided in a scientific paper. In a scientific context, this is an hypothesis. Do you have any mean to verify or support it ?

P19L11-12: It is difficult to understand why when the RCM is strongly influenced by the boundary conditions of ERA-Interim, it can even outperform it.

P19L13-14: The improvement of the IWV diurnal cycle due to an upgraded microphysical scheme is not obvious to me. It is also not clear to which seasons you are referring to.

P20: Again in the conclusions the various explanations given along the paper are far from being convincing.

P21L1-2: Why at 20 km the feedbacks between water vapour and other meteorological variables "could lack sufficient representation" ? Which processes to you have in mind ?

P21L4-5: I am not sure to understand what the proposed methodology is. How would you perform assimilation of GNSS data since the ALARO-SURFEX model is run in climate mode ?

P21L7: Add a reference to ERA5

I have many additional minor changes (mostly rephrasing for clarifications) that I do not find necessary to include at this stage. A recommendation would be to have the paper corrected by a native English speaking person.

References :

Davis, J. L., T. A. Herring, I. I. Shapiro, A. E. Rogers, and G. Elgered, 1985 : Geodesy by radio interferometry : Effects of atmospheric modeling errors on estimates of baseline length. Radio Science, 20, 1593-1607

Seity, Y., P. Brousseau, S. Malardel, G. Hello, P. Bénard, F. Bouttier, C. Lac, and V. Masson, 201 : The AROME-France Convective-Scale Operational Model. Mon. Wea. Rev., 139, 976–991, https://doi.org/10.1175/2010MWR3425.1